# PARALLEL NEURAL TEXT-TO-SPEECH

## ABSTRACT

In this work, we first propose ParaNet, a non-autoregressive *seq2seq* model that converts text to spectrogram. It is fully convolutional and obtains $46.7\times$ speed-up over Deep Voice 3 (Ping et al., 2018b) at synthesis while maintaining comparable speech quality using a WaveNet vocoder. ParaNet also produces stable alignment between text and speech on the challenging test sentences by iteratively improving the attention in a layer-by-layer manner. Based on ParaNet, we build the first fully parallel neural text-to-speech system using parallel neural vocoders, which can synthesize speech from text through a single feed-forward pass. We investigate several parallel vocoders within the TTS system, including variants of IAF vocoders and bipartite flow vocoder. [1]

## 1 INTRODUCTION

Text-to-speech (TTS), also called speech synthesis, has long been a vital tool in a variety of applications, such as human-computer interactions, virtual assistant, and content creation. Traditional TTS systems are based on multi-stage hand-engineered pipelines (Taylor, 2009). In recent years, deep neural networks based autoregressive models have attained state-of-the-art results, including high-fidelity audio synthesis (van den Oord et al., 2016), and much simpler sequence-to-sequence (*seq2seq*) pipelines (Sotelo et al., 2017; Wang et al., 2017; Ping et al., 2018b). In particular, one of the most popular neural TTS pipelines consists of two components (Ping et al., 2018b; Shen et al., 2018): (i) an autoregressive *seq2seq* model that generates mel spectrogram from text, and (ii) an autoregressive neural vocoder (e.g., WaveNet) that generates raw waveform from mel spectrogram. This pipeline requires much less expert knowledge and only needs pairs of audio and transcript as training data.

The autoregressive nature of these components leads to slow synthesis, because they operate sequentially at a high temporal resolution of waveform samples and spectrogram. Most recently, Parallel WaveNet (van den Oord et al., 2018) and ClariNet (Ping et al., 2018a) were proposed for parallel waveform synthesis by distilling an inverse autoregressive flow (IAF) (Kingma et al., 2016) from a pretrained WaveNet. However, they still rely on autoregressive or recurrent components to predict the frame-level acoustic features (e.g., 100 frames per second), which can be slow at synthesis on modern hardware optimized for parallel execution.

In this work, we introduce a fully parallel neural TTS system by proposing a non-autoregressive text-to-spectrogram model. Our major contributions are as follows:

1. We propose ParaNet, the first non-autoregressive attention-based architecture for TTS, which is fully convolutional and converts text to mel spectrogram. We compare the non-autoregressive ParaNet with its autoregressive counterpart (Ping et al., 2018b) in terms of speech quality, synthesis speed and attention stability. It achieves $\sim 46.7$ times speed-up over Deep Voice 3 (DV3) at synthesis, while maintaining comparable speech quality using a WaveNet vocoder. ParaNet iteratively refines the attention alignment between text and spectrogram in a layer-by-layer manner, and it can produce stable attentions on the challenging test sentences as DV3 with attention masking.

2. Based on ParaNet, we build the first fully parallel neural TTS system by combining ParaNet with parallel neural vocoders. Our system can generate speech from text through a single feed-forward pass. We investigate several parallel vocoders within the parallel TTS system, including the distilled IAF vocoder (Ping et al., 2018a) and WaveGlow (Prenger et al., 2019).

---

[1] Synthesized speech samples are in: `https://parallel-neural-tts-demo.github.io/`.

To explore the possibility of training IAF vocoder *without distillation* in this parallel system, we also propose an alternative approach, WaveVAE, which can be trained from scratch within the variational autoencoder (VAE) framework (Kingma and Welling, 2014).

We organize the rest of this paper as follows. Section 2 discusses related work. We introduce the non-autoregressive text-to-spectrogram architecture in Section 3, and discuss the parallel neural vocoders in Section 4. We report experimental results in Section 5, and conclude the paper in Section 6.

## 2 RELATED WORK

Neural speech synthesis has obtained the state-of-the-art results and gained a lot of attention. Several neural TTS systems were proposed, including WaveNet (van den Oord et al., 2016), Deep Voice (Arık et al., 2017a), Deep Voice 2 (Arık et al., 2017b), Deep Voice 3 (Ping et al., 2018b), Tacotron (Wang et al., 2017), Tacotron 2 (Shen et al., 2018), Char2Wav (Sotelo et al., 2017), VoiceLoop (Taigman et al., 2018), WaveRNN (Kalchbrenner et al., 2018), Transformer TTS (Li et al., 2019) and ClariNet (Ping et al., 2018a). In particular, Tacotron and Deep Voice 3 employ *seq2seq* framework with the attention mechanism (Bahdanau et al., 2015), yielding much simpler pipeline compared to traditional multi-stage pipeline. Their excellent extensibility leads to promising results for several challenging tasks, such as voice cloning (Arik et al., 2018; Nachmani et al., 2018; Jia et al., 2018; Chen et al., 2019). All of these state-of-the-art TTS systems are based on autoregressive models.

RNN-based autoregressive models, such as Tacotron and WaveRNN (Kalchbrenner et al., 2018), lack parallelism at both training and synthesis. CNN-based autoregressive models, such as Deep Voice 3 and WaveNet, enable parallel processing at training, but they still operate sequentially at synthesis since each output element must be generated before it can be passed in as input at the next time step. Recently, there are some non-autoregressive models proposed for neural machine translation. For example, Gu et al. (2018) trains a feed-forward neural network conditioned on fertility values, which is obtained from an external alignment system. Kaiser et al. (2018) proposes a latent variable model for fast decoding, while it remains autoregressive between latent variables. Lee et al. (2018) iteratively refines the output sequence through a denoising autoencoder framework. Arguably, non-autoregressive model plays a more important role in text-to-speech, where the output speech spectrogram consists of hundreds of time steps for a short text with a few words. To the best of our knowledge, our work is the first non-autoregressive *seq2seq* model for TTS and provides as much as 46.7 times speed-up at synthesis over its convolutional autoregressive counterpart.

Flow-based generative models (Rezende and Mohamed, 2015; Kingma et al., 2016; Dinh et al., 2017; Kingma and Dhariwal, 2018) transform a simple initial distribution into a more complex one by applying a series of invertible transformations. In previous work, flow-based models have obtained state-of-the-art results for parallel waveform synthesis (van den Oord et al., 2018; Ping et al., 2018a; Prenger et al., 2019; Kim et al., 2019; Yamamoto et al., 2019).

Variational autoencoder (VAE) (Kingma and Welling, 2014; Rezende et al., 2014) has been applied for representation learning of natural speech for years. It models either the generative process of waveform samples (Chung et al., 2015; van den Oord et al., 2017), or spectrograms (Hsu et al., 2019). In previous work, autoregressive or recurrent neural networks are employed as the decoder of VAE (Chung et al., 2015; van den Oord et al., 2017), but they can be quite slow at synthesis.

## 3 TEXT-TO-SPECTROGRAM MODEL

Our parallel TTS system has two components: 1) a feed-forward text-to-spectrogram model, and 2) a parallel waveform synthesizer conditioned on spectrogram. In this section, we first present an autoregressive text-to-spectrogram model derived from Deep Voice 3 (DV3) (Ping et al., 2018b). We then introduce ParaNet, a non-autoregressive text-to-spectrogram model (see Figure 1).

### 3.1 AUTOREGRESSIVE ARCHITECTURE

Our autoregressive model is based on DV3, a convolutional text-to-spectrogram model, which consists of three components:

- **Encoder**: A convolutional encoder, which encodes text into internal hidden representation.

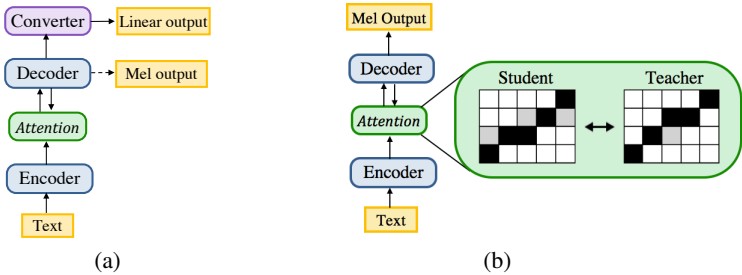

Figure 1: (a) Autoregressive text-to-spectrogram model. The dashed line depicts the autoregressive decoding of mel spectrogram at inference. (b) Non-autoregressive ParaNet model, which distills the attention from a pretrained autoregressive model.

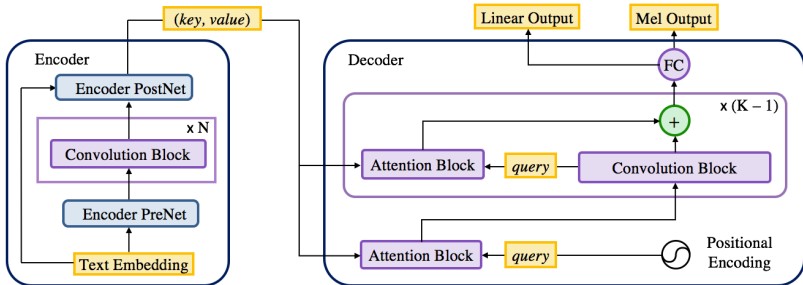

Figure 2: Architecture of ParaNet. Its encoder provides *key* and *value* as the textual representation. The first attention block in decoder gets positional encoding as the *query* and is followed by non-causal convolution blocks and attention blocks.

- **Decoder**: A *causal* convolutional decoder, which decodes the encoder representation with an *attention* mechanism to log-mel spectragrams in an *autoregressive* manner with an $\ell_1$ loss. It starts with $1 \times 1$ convolutions to preprocess the input log-mel spectrograms.
- **Converter**: A *non-causal* convolutional post-processing network, which processes the hidden representation from the decoder using both past and future context information and predicts the log-linear spectrograms with an $\ell_1$ loss. It enables bidirectional processing.

All these components use the same 1-D convolution with a gated linear unit as in DV3. The major difference between our model and DV3 is the decoder architecture. The decoder of DV3 has multiple attention-based layers, where each layer consists of a causal convolution block followed by an attention block. To simplify the *attention distillation* described in Section 3.3.1, our autoregressive decoder has only one attention block at its first layer. We find that reducing the number of attention blocks does not hurt the generated speech quality in general.

## 3.2 NON-AUTOREGRESSIVE ARCHITECTURE

Our ParaNet (see Figure 2) uses the same encoder architecture as the autoregressive model. The decoder of ParaNet, conditioned solely on the hidden representation from the encoder, predicts the entire sequence of log-mel spectrograms in a feed-forward manner. As a result, both its training and synthesis can be performed in parallel. Specially, we make the following major architecture modifications from the autoregressive *seq2seq* model to the non-autoregressive model:

1. **Non-autoregressive decoder**: Without the autoregressive generative constraint, the decoder can use *non-causal* convolution blocks to take advantage of future context information and to improve model performance. In addition to log-mel spectrograms, it also predicts log-linear spectrograms with an $\ell_1$ loss for slightly better performance.
2. **No converter**: Non-autoregressive model removes the *non-causal* converter since it already employs a *non-causal* decoder. Note that, the major motivation of introducing non-causal *converter* in DV3 is to refine the decoder predictions based on bidirectional context information provided by non-causal convolutions (Ping et al., 2018b).

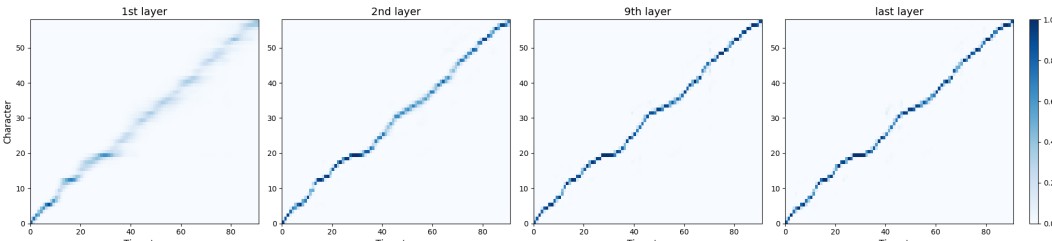

Figure 3: Our ParaNet iteratively refines the attention alignment in a layer-by-layer way. One can see the 1st layer attention is mostly dominated by the positional encoding prior. It becomes more and more confident about the alignment in the subsequent layers.

### 3.3 PARALLEL ATTENTION MECHANISM

It is challenging for the non-autoregressive model to learn the accurate alignment between the input text and output spectrogram. Note that, we need the full parallelism within the attention mechanism. For example, the location-sensitive attention (Chorowski et al., 2015; Shen et al., 2018) improves attention stability, but it performs sequentially at both training and synthesis, because it uses the cumulative attention weights from previous decoder time steps as an additional feature. Previous non-autoregressive decoders rely on an external alignment system (Gu et al., 2018), or an autoregressive latent variable model (Kaiser et al., 2018). In this work, we present several simple & effective techniques, which could obtain accurate and stable alignment with the multi-step attention (Gehring et al., 2017). In particular, our non-autoregressive decoder can iteratively refine the attention alignment between text and mel spectrogram in a layer-by-layer manner as illustrated in Figure 3. Specially, the decoder adopts a dot-product attention mechanism and consists of $K$ attention blocks (see Figure 2), where each attention block uses the per-time-step *query* vectors from convolution block and per-time-step *key* vectors from encoder to compute the attention weights (Ping et al., 2018b). The attention block computes *context* vectors as the weighted average of the *value* vectors from the encoder. The non-autoregressive decoder starts with an attention block, in which the *query* vectors are solely positional encoding (see Section 3.3.2 for details). The first attention block then provides the input for the convolution block at the next attention-based layer.

#### 3.3.1 ATTENTION DISTILLATION

We use the attention alignments from the pretrained autoregressive model to guide the training of non-autoregressive ParaNet. Specially, we minimize the cross entropy between their attention distributions. We denote the attention weights from ParaNet as $W_{i,j}^{(k)}$, where $i$ and $j$ index the time-step of encoder and decoder respectively, and $k$ refers to the $k$-th attention block within the decoder. Note that, the attention weights $\{W_{i,j}^{(k)}\}_{i=1}^{M}$ form a valid distribution. We compute the *attention loss* as the average cross entropy between the student and teacher's attention distributions:

$$l_{\text{atten}} = -\frac{1}{KN}\sum_{k=1}^{K}\sum_{j=1}^{N}\sum_{i=1}^{M}W_{i,j}^{T}\log W_{i,j}^{(k)}, \tag{1}$$

where $W_{i,j}^{T}$ are the attention weights from the autoregressive teacher, $M$ and $N$ are the lengths of encoder and decoder, respectively. Our final loss function is a linear combination of $l_{\text{atten}}$ and $\ell_1$ losses from spectrogram predictions. We set the coefficient of $l_{\text{atten}}$ as 4, and other coefficients as 1.

#### 3.3.2 POSITIONAL ENCODING

We use a similar positional encoding as in DV3 at every attention block. The positional encoding is added to both *key* and *query* vectors in the attention block, which forms an inductive bias for monotonic attention. Note that, ParaNet solely relies on its attention mechanism to decode mel spectrograms from the encoded textual features, without any autoregressive input. This makes the positional encoding even more crucial in guiding the attention to follow a monotonic progression over time at training. The positional encodings are set as $h_p(i, k) = \sin\left(\omega_s i / 10000^{k/d}\right)$ (for even $i$), and $\cos\left(\omega_s i / 10000^{k/d}\right)$ (for odd $i$), where $i$ is the time-step index, $k$ is the channel index, $d$ is the total number of channels in the positional encoding, and $\omega_s$ is the *position rate* which indicates the average

slope of the line in the attention distribution and roughly corresponds to the speed of speech. We set $\omega_s$ in the following ways:

- For the autoregressive teacher, $\omega_s$ is set to one for the positional encoding of *query*. For the *key*, it is set to the averaged ratio of the time-steps of spectrograms to the time-steps of textual features, which is around 6.3 across our training dataset. Taking into account that a reduction factor of 4 is used to simplify the learning of attention mechanism (Wang et al., 2017; Ping et al., 2018b) , $\omega_s$ is simply set as $6.3/4$ for the *key* at both training and synthesis.
- For ParaNet, $\omega_s$ is also set to one for the *query*, while $\omega_s$ for the *key* is calculated differently. At training, $\omega_s$ is set to the ratio of the lengths of spectrograms and text for each individual training instance, which is also divided by a reduction factor of 4. At synthesis, we need to specify the length of output spectrogram and the corresponding $\omega_s$, which actually controls the speech rate of the generated audios (see Section III on demo website). F, we simply set $\omega_s$ to be $6.3/4$ as in autoregressive model, and the length of output spectrogram as $6.3/4$ times the length of input text. Such a setup yields an initial attention in the form of a diagonal line and guides the non-autoregressive decoder to refine its attention layer by layer (see Figure 3).

### 3.3.3 ATTENTION MASKING

Inspired by the attention masking in DV3 (Ping et al., 2018b), we propose a different attention masking scheme for ParaNet at synthesis. For each *query* from decoder, instead of computing the softmax over the entire set of encoder *key* vectors, we compute the softmax only over a fixed window centered around the *target position* and going forward and backward several time-steps (e.g., 3). The *target position* is calculated as $\lfloor i_{query} \times 4/6.3 \rfloor$, where $i_{query}$ is the time-step index of the *query* vector, and $\lfloor \rceil$ is the rounding operator. We observe that this strategy reduces serious attention errors such as repeating or skipping words, and also yields clearer pronunciations, thanks to its more condensed attention distribution. Note that, this attention masking is shared across all attention blocks once it is generated, and does not prevent the parallel synthesis of the non-autoregressive model.

## 4 PARALLEL NEURAL VOCODER

As an indispensable component in our parallel neural TTS system, the parallel neural vocoder converts the mel spectrogram returned from the non-autoregressive ParaNet into the raw waveform. In this section, we discuss several existing neural vocoders, and explore a new alternative in the system.

### 4.1 IAF VOCODER VS. BIPARTITE FLOW VOCODER

Inverse autoregressive flow (IAF) (Kingma et al., 2016) is a special type of normalizing flow where each invertible transformation is based on an autoregressive neural network. IAF performs synthesis in parallel and can easily reuse the expressive autoregressive architecture, such as WaveNet (van den Oord et al., 2016), which leads to the state-of-the-art results for speech synthesis (van den Oord et al., 2018; Ping et al., 2018a). However, the likelihood evaluation in IAF is autoregressive and slow, thus previous training methods rely on probability density distillation from a pretrained autoregressive WaveNet (van den Oord et al., 2018; Ping et al., 2018a). The two-stage distillation process complicates the training pipeline and may even introduce pathological optimization (Huang et al., 2019). RealNVP (Dinh et al., 2017) and Glow (Kingma and Dhariwal, 2018) are different types of normalizing flows, where both synthesis and likelihood evaluation can be performed in parallel by enforcing bipartite architecture constraints. Most recently, both methods were applied as parallel neural vocoders and can be trained from scratch (Prenger et al., 2019; Kim et al., 2019). However, these models are less expressive than their autoregressive and IAF counterparts, because half of the variables are unchanged after each transformation. In general, these bipartite flows require larger number of layers and hidden units, which lead to huge number of parameters. For example, a WaveGlow vocoder (Prenger et al., 2019) has 87.88M parameters, whereas IAF vocoder has much smaller footprint with only 2.15M parameters (Ping et al., 2018a), making it more preferred in production deployment.

### 4.2 WAVEVAE

Given a small-footprint IAF vocoder, it is interesting to investigate whether it can be trained *without* the density distillation (e.g., Huang et al., 2019). Our method uses the VAE framework, thus it is

termed as WaveVAE. In contrast to van den Oord et al. (2018); Ping et al. (2018a), WaveVAE can be trained from scratch by jointly optimizing the encoder $q_\phi(\boldsymbol{z}|\boldsymbol{x}, \boldsymbol{c})$ and decoder $p_\theta(\boldsymbol{x}|\boldsymbol{z}, \boldsymbol{c})$, where $\boldsymbol{z}$ is latent variables and $\boldsymbol{c}$ is the mel spectrogram conditioner. We omit $\boldsymbol{c}$ for concise notation hereafter.

**Encoder:** The encoder of WaveVAE $q_\phi(\boldsymbol{z}|\boldsymbol{x})$ is parameterized by a Gaussian WaveNet (Ping et al., 2018a) that maps the ground truth audio $\boldsymbol{x}$ into the latent representation $\boldsymbol{z}$ with the same length. Specifically, the Gaussian WaveNet models $x_t$ given the previous samples $x_{<t}$ as $x_t \sim \mathcal{N}\big(\mu(x_{<t}; \phi), \sigma(x_{<t}; \phi)\big)$, where the mean $\mu(x_{<t}; \phi)$ and scale $\sigma(x_{<t}; \phi)$ are predicted by the WaveNet, respectively. The encoder posterior is constructed as,

$$q_\phi(\boldsymbol{z}|\boldsymbol{x}) = \prod_t q_\phi(z_t \mid x_{\leq t}), \quad \text{where} \quad q_\phi(z_t \mid x_{\leq t}) = \mathcal{N}\big(\frac{x_t - \mu(x_{<t}; \phi)}{\sigma(x_{<t}; \phi)}, \varepsilon\big). \tag{2}$$

Note that, the mean $\mu(x_{<t}; \phi)$ and scale $\sigma(x_{<t}; \phi)$ are applied for "whitening" the posterior distribution. We introduce a trainable scalar $\varepsilon > 0$ to capture the global variation, which will ease the optimization process. Given the observed $\boldsymbol{x}$, the $q_\phi(\boldsymbol{z}|\boldsymbol{x})$ admits parallel sampling of latents $\boldsymbol{z}$. Note that, there is a connection between the encoder of WaveVAE and the teacher model of ClariNet, as both of them use a Gaussian WaveNet to guide the training of the inverse autoregressive flow (IAF) (Kingma et al., 2016) for parallel wave generation.

**Decoder:** Our decoder $p_\theta(\boldsymbol{x}|\boldsymbol{z})$ is is parameterized by the one-step-ahead predictions from an IAF (Ping et al., 2018a). Specially, we let $\boldsymbol{z}^{(0)} = \boldsymbol{z}$ and apply a stack of IAF transformations from $\boldsymbol{z}^{(0)} \rightarrow \ldots \boldsymbol{z}^{(i)} \rightarrow \ldots \boldsymbol{z}^{(n)}$, and each transformation $\boldsymbol{z}^{(i)} = f(\boldsymbol{z}^{(i-1)}; \theta)$ is defined as,

$$\boldsymbol{z}^{(i)} = \boldsymbol{z}^{(i-1)} \cdot \boldsymbol{\sigma}^{(i)} + \boldsymbol{\mu}^{(i)}, \tag{3}$$

where $\mu_t^{(i)} = \mu(z_{<t}^{(i-1)}; \theta)$ and $\sigma_t^{(i)} = \sigma(z_{<t}^{(i-1)}; \theta)$ are shifting and scaling variables modeled by a Gaussian WaveNet. As a result, given $\boldsymbol{z}^{(0)} \sim \mathcal{N}(\boldsymbol{\mu}^{(0)}, \boldsymbol{\sigma}^{(0)})$ from the Gaussian prior or encoder, the one-step-ahead prediction $p(z_t^{(n)} \mid z_{<t}^{(0)})$ also follows Gaussian with scale and mean as,

$$\boldsymbol{\sigma}^{\text{tot}} = \prod_{i=0}^n \boldsymbol{\sigma}^{(i)}, \quad \boldsymbol{\mu}^{\text{tot}} = \sum_{i=0}^n \boldsymbol{\mu}^{(i)} \prod_{j>i}^n \boldsymbol{\sigma}^{(j)}. \tag{4}$$

Lastly, we set $\boldsymbol{x} = \boldsymbol{\epsilon} \cdot \boldsymbol{\sigma}^{\text{tot}} + \boldsymbol{\mu}^{\text{tot}}$, where $\boldsymbol{\epsilon} \sim \mathcal{N}(0, I)$. For the generative process, we use the standard Gaussian prior $p(\boldsymbol{z}) = \mathcal{N}(0, I)$.

**VAE objective:** We maximize the evidence lower bound (ELBO) for observed $\boldsymbol{x}$ in VAE,

$$\max_{\phi, \theta} \mathbb{E}_{q_\phi(\boldsymbol{z}|\boldsymbol{x})}\big[\log p_\theta(\boldsymbol{x}|\boldsymbol{z})\big] - \text{KL}\big(q_\phi(\boldsymbol{z}|\boldsymbol{x}) \,\|\, p(\boldsymbol{z})\big), \tag{5}$$

where the KL divergence can be calculated in closed-form as both $q_\phi(\boldsymbol{z}|\boldsymbol{x})$ and $p(\boldsymbol{z})$ are Gaussians,

$$\text{KL}\big(q_\phi(\boldsymbol{z}|\boldsymbol{x}) \,\|\, p(\boldsymbol{z})\big) = \sum_t \log \frac{1}{\varepsilon} + \frac{1}{2}\Big(\varepsilon^2 - 1 + \big(\frac{x_t - \mu(x_{<t})}{\sigma(x_{<t})}\big)^2\Big). \tag{6}$$

The reconstruction term in Eq. (5) is intractable to compute exactly. We approximate this term via stochastic optimization, by drawing a sample $\boldsymbol{z}$ from the encoder $q_\phi(\boldsymbol{z}|\boldsymbol{x})$ through the reparameterization trick, and evaluating the likelihood $\log p_\theta(\boldsymbol{x}|\boldsymbol{z})$. Although $\boldsymbol{z}$ has the same size as $\boldsymbol{x}$, it will be disentangled by minimizing this KL divergence. To avoid the "posterior collapse", in which the posterior distribution $q_\phi(\boldsymbol{z}|\boldsymbol{x})$ quickly collapses to the white noise prior $p(\boldsymbol{z})$ at the early stage of training, we apply the annealing strategy for KL divergence, where its weight is gradually increased from 0 to 1, via a sigmoid function (Bowman et al., 2016). Through it, the encoder can encode sufficient information into the latent representations at the early stage of training, and then gradually regularize the latent representation by increasing the weight of the KL divergence.

**STFT loss:** Similar to Ping et al. (2018a), we also add a short-term Fourier transform (STFT) based loss to improve the quality of the synthesized speech. We define the *STFT loss* as the summation of an $\ell_2$ loss on the magnitudes of STFT and an $\ell_1$ loss on the log-magnitudes of STFT between the output audio and ground truth audio (Arık et al., 2019; Wang et al., 2019). For STFT, we set the frame-shift to 12.5ms, Hanning window length to 50ms, and the FFT size to 2048. We consider two STFT losses in our objective: $(i)$ the STFT loss between ground truth audio and reconstructed audio using encoder $q_\phi(\boldsymbol{z}|\boldsymbol{x})$; $(ii)$ the STFT loss between ground truth audio and synthesized audio using the prior $p(\boldsymbol{z})$, with the purpose of reducing the gap between reconstruction and synthesis. Our final loss is a linear combination of the terms in Eq. (5) and the STFT losses. The corresponding coefficients are simply set to be one in all of our experiments.

Table 1: Mean Opinion Score (MOS) ratings with 95% confidence intervals for comparison.

| Model | MOS score |
|---|---|
| Deep Voice 3 + WaveNet | $4.09 \pm 0.26$ |
| ParaNet + WaveNet | $4.01 \pm 0.24$ |

## 5 EXPERIMENTS

In this section, we first compare text-to-spectrogram ParaNet with its autoregressive counterpart DV3 in terms of speech quality, synthesis speed, and attention stability. We then present the results of parallel neural TTS system paired with various parallel neural vocoders. In our experiment, we use a proprietary English speech dataset containing about 20 hours of speech data from a female speaker with a sampling rate of 48 kHz. We downsample the audios to 24 kHz.

### 5.1 PARANET VS. DEEP VOICE 3

For both ParaNet and DV3, we use the mixed representation of characters and phonemes (Ping et al., 2018b). The default hyperparameters of ParaNet and DV3 are provided in Appendix A. Both ParaNet and DV3 are trained for 1M steps using Adam optimizer (Kingma and Ba, 2015) with batch size 16. We find that larger kernel width and deeper layers generally help improve the performance of ParaNet. Our best non-autoregressive ParaNet (17.61M) is 2.57 times larger than the autoregressive DV3 (6.85M) in terms of the number of parameters, but it obtains significant speedup at synthesis.

**Speech quality:** We compare the speech quality of ParaNet and DV3 using WaveNet vocoder. We train 20-layer WaveNets with residual channel 256 conditioned on the predicted mel spectrogram from ParaNet and DV3, respectively. We apply two layers of convolution block to process the predicted mel spectrogram, and use two layers of transposed 2-D convolution (in time and frequency) interleaved with leaky ReLU ($\alpha = 0.4$) to upsample the outputs from frame-level to sample-level. We use the crowdMOS toolkit (Ribeiro et al., 2011) for subjective Mean Opinion Score (MOS) evaluation. We report the MOS results in Table 1. The non-autoregressive ParaNet can provide comparable quality of speech as the autoregressive DV3 using WaveNet vocoder.

**Speedup at synthesis:** We compare the largest ParaNet with the autoregressive DV3 in terms of inference latency. We construct a custom 15-sentence test set (see Appendix D) and run inference for 50 runs on each of the 15 sentences (batch size is set to 1). The average inference latencies over 50 runs and 15 sentences are $0.024$ and $1.12$ seconds on NVIDIA GeForce GTX 1080 Ti for the non-autoregressive and autoregressive models, respectively. Hence, our ParaNet brings about $46.7$ times speed-up compared to its autoregressive counterpart at synthesis.

**Attention error analysis:** In autoregressive models, there is a noticeable discrepancy between the teacher-forced training and autoregressive inference, which can yield accumulated errors along the generated sequence at synthesis (e.g., Bengio et al., 2015). In neural TTS, this discrepancy leads to miserable attention errors at autoregressive inference, including (i) repeated words, (ii) mispronunciations, and (iii) skipped words (see Ping et al. (2018b) for detailed examples), which is a critical problem for online deployment of attention-based neural TTS systems. We perform an attention error analysis for our non-autoregressive ParaNet on a 100-sentence test set (see Appendix C), which includes particularly-challenging cases from deployed TTS systems (e.g. dates, acronyms, URLs, repeated words, proper nouns, foreign words etc.). In Table 2, we find that the non-autoregressive ParaNet has much fewer attention errors than its autoregressive counterpart at synthesis (12 vs. 37) without attention mask. Although our ParaNet distills the (teacher-forced) attentions from an autoregressive model, it only takes textual inputs at both training and synthesis and does not have the similar discrepancy as in autoregressive model. In previous work, attention masking was applied to enforce the monotonic attentions and reduce attention errors, and was demonstrated to be effective in Deep Voice 3 (Ping et al., 2018b). We find that our non-autoregressive ParaNet has slightly fewer attention errors than autoregressive DV3 (6 vs. 8), when both of them use the attention masking.

**Ablation Study:** We perform ablation studies to verify the effectiveness of several techniques used in ParaNet, including attention distillation, positional encoding, and stacking decoder layers to refine the attention alignment in a layer-by-layer manner. We evaluate the performance of a non-autoregressive ParaNet model trained without attention distillation and find that it fails to learn meaningful attention alignment. The synthesized audios are unintelligible and mostly pure noise. Similarly, we train

Table 2: Attention error counts for text-to-spectrogram models on the 100-sentence test set. One or more mispronunciations, skips, and repeats count as a single mistake per utterance. The non-autoregressive ParaNet (17-layer decoder) with attention mask obtains the fewest attention errors in total. For ablation study, we include the results for two additional ParaNet models. They have 6 and 12 decoder layers and are denoted as ParaNet-6 and ParaNet-12, respectively.

| Model | Attention mask | Repeat | Mispronounce | Skip | Total |
|---|---|---|---|---|---|
| Deep Voice 3 | | 12 | 10 | 15 | 37 |
| **ParaNet** | No | **1** | **4** | **7** | **12** |
| ParaNet-12 | | 5 | 7 | 5 | 17 |
| ParaNet-6 | | 4 | 11 | 11 | 26 |
| Deep Voice 3 | | 1 | 4 | 3 | 8 |
| **ParaNet** | Yes | **2** | **4** | **0** | **6** |
| ParaNet-12 | | 4 | 6 | 2 | 12 |
| ParaNet-6 | | 3 | 10 | 3 | 16 |

Table 3: Mean Opinion Score (MOS) ratings with 95% confidence intervals for comparison.

| Neural TTS system | # parameters | MOS score |
|---|---|---|
| Deep Voice 3 + IAF (distilled) | 8.99 M | $3.93 \pm 0.27$ |
| Deep Voice 3 + IAF (WaveVAE) | 8.99 M | $3.70 \pm 0.29$ |
| Deep Voice 3 + WaveGlow | 94.73 M | $3.96 \pm 0.25$ |
| ParaNet + IAF (distilled) | 19.76 M | $3.52 \pm 0.28$ |
| ParaNet + IAF (WaveVAE) | 19.76 M | $3.25 \pm 0.34$ |
| ParaNet + WaveGlow | 105.49 M | $3.21 \pm 0.31$ |

another non-autoregressive ParaNet model without adding positional encoding in the attention block. The resulting model only learns very blurry attention alignment and cannot synthesize intelligible speech. Finally, we train two non-autoregressive ParaNet models with 6 and 12 decoder layers, respectively, and compare them with the default non-autoregressive ParaNet model which has 17 decoder layers (denoted as ParaNet in Table 2). We conduct the same attention error analysis on the 100-sentence test set and the results are shown in Table 2. We find that increasing the number of decoder layers for non-autoregressive ParaNet can reduce the total number of attention errors, in both cases with and without applying attention mask at synthesis.

## 5.2 TTS SYSTEM WITH PARALLEL VOCODERS

We evaluate the neural TTS systems with parallel neural vocoders, including distilled IAF vocoder (Ping et al., 2018a), the proposed WaveVAE, and WaveGlow (Prenger et al., 2019). The experimental setup can be found in Appendix B. WaveGlow produces high-frequency artifacts for some utterances. To remedy this, we applied the denoising function with strength 0.1 in the repository. [2] It is effective when the predicted mel spectrograms are from DV3, but not quite effective when the predicted mel spectrograms are from ParaNet. We report the MOS evaluation results of TTS systems in Table 3. Note that, all of parallel neural vocoders are trained on ground-truth mel spectrograms. We find training them on predicted mel spectrograms does not stably improve the audio quality, which is different from using the WaveNet vocoder. The distilled IAF vocoder tends to be more robust than WaveVAE and WaveGlow in the parallel TTS system. Although both WaveVAE and WaveGlow can be trained from scratch, WaveVAE requires much fewer parameters than WaveGlow.

## 6 CONCLUSION

In this work, we build a fully parallel neural TTS system by proposing a non-autoregressive text-to-spectrogram model and combining it with several parallel vocoders. Our non-autoregressive text-to-spectrogram model obtains $46.7\times$ speed-up over Deep Voice 3. In addition, we investigate various parallel vocoders in this TTS system. To explore the possibility of training IAF vocoder without distillation for parallel waveform synthesis, we propose an alternative approach, WaveVAE, which avoids the need for training a separate autoregressive WaveNet, and can be trained from scratch.

---

[2] https://github.com/NVIDIA/waveglow

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

# Appendices

## A  HYPERPARAMETERS FOR TEXT-TO-SPECTROGRAM MODEL

Table 4: Hyperparameters of autoregressive seq2seq model and non-autoregressive seq2seq model in the experiment.

| Hyperparameter | Autoregressive Model | Non-autoregressive Model |
|---|---|---|
| FFT Size | 2048 | 2048 |
| FFT Window Size / Shift | 1200 / 300 | 1200 / 300 |
| Audio Sample Rate | 24000 | 24000 |
| Reduction Factor $r$ | 4 | 4 |
| Mel Bands | 80 | 80 |
| Character Embedding Dim. | 256 | 256 |
| Encoder Layers / Conv. Width / Channels | 7 / 5 / 64 | 7 / 9 / 64 |
| Decoder PreNet Affine Size | 128, 256 | N/A |
| Decoder Layers / Conv. Width | 4 / 5 | 17 / 7 |
| Attention Hidden Size | 128 | 128 |
| Position Weight / Initial Rate | 1.0 / 6.3 | 1.0 / 6.3 |
| PostNet Layers / Conv. Width / Channels | 5 / 5 / 256 | N/A |
| Dropout Keep Probability | 0.95 | 1.0 |
| ADAM Learning Rate | 0.001 | 0.001 |
| Batch Size | 16 | 16 |
| Max Gradient Norm | 100 | 100 |
| Gradient Clipping Max. Value | 5.0 | 5.0 |
| Total Number of Parameters | 6.85M | 17.61M |

## B  DETAILS OF PARALLEL NEURAL VOCODERS

For the distilled IAF vocoder, we use the same architecture as ClariNet (Ping et al., 2018a). It consists of four stacked Gaussian IAF blocks, which are parameterized by [10, 10, 10, 30]-layer WaveNets respectively, with 64 residual & skip channels and filter size 3 in dilated convolutions. The IAF is conditioned on log-mel spectrograms with two layers of transposed 2-D convolution as in ClariNet. We use the same teacher-student setup for ClariNet as in Ping et al. (2018a) and we train a 20-layer Gaussian autoregressive WaveNet as the teacher model. For the encoder in WaveVAE, we also use a 20-layers Gaussian WaveNet conditioned on log-mel spectrograms. For the decoder, we use the same architecture as the distilled IAF. Both the encoder and decoder of WaveVAE share the same conditioner network. We use Adam optimizer with 1000K steps for distilled IAF. For WaveVAE, we train it for 400K because it converges much faster. The learning rate is set to 0.001 at the beginning and annealed by half for every 200K steps for both models. For WaveGlow, we use the open source implementation with default hyperparameters [3], except sampling rate (22.05kHz→24kHz), FFT window length (1024→1200), and FFT window shift (256→300) for handling the 24kHz dataset. It is trained for 1M steps.

## C  100-SENTENCE TEST SET

The 100 sentences used to quantify the results in Table 1 are listed below (note that % corresponds to pause):

1. A B C%.
2. X Y Z%.
3. HURRY%.
4. WAREHOUSE%.

---

[3] https://github.com/NVIDIA/waveglow

5. REFERENDUM%.
6. IS IT FREE%?
7. JUSTIFIABLE%.
8. ENVIRONMENT%.
9. A DEBT RUNS%.
10. GRAVITATIONAL%.
11. CARDBOARD FILM%.
12. PERSON THINKING%.
13. PREPARED KILLER%.
14. AIRCRAFT TORTURE%.
15. ALLERGIC TROUSER%.
16. STRATEGIC CONDUCT%.
17. WORRYING LITERATURE%.
18. CHRISTMAS IS COMING%.
19. A PET DILEMMA THINKS%.
20. HOW WAS THE MATH TEST%?
21. GOOD TO THE LAST DROP%.
22. AN M B A AGENT LISTENS%.
23. A COMPROMISE DISAPPEARS%.
24. AN AXIS OF X Y OR Z FREEZES%.
25. SHE DID HER BEST TO HELP HIM%.
26. A BACKBONE CONTESTS THE CHAOS%.
27. TWO A GREATER THAN TWO N NINE%.
28. DON'T STEP ON THE BROKEN GLASS%.
29. A DAMNED FLIPS INTO THE PATIENT%.
30. A TRADE PURGES WITHIN THE B B C%.
31. I'D RATHER BE A BIRD THAN A FISH%.
32. I HEAR THAT NANCY IS VERY PRETTY%.
33. I WANT MORE DETAILED INFORMATION%.
34. PLEASE WAIT OUTSIDE OF THE HOUSE%.
35. N A S A EXPOSURE TUNES THE WAFFLE%.
36. A MIST DICTATES WITHIN THE MONSTER%.
37. A SKETCH ROPES THE MIDDLE CEREMONY%.
38. EVERY FAREWELL EXPLODES THE CAREER%.
39. SHE FOLDED HER HANDKERCHIEF NEATLY%.
40. AGAINST THE STEAM CHOOSES THE STUDIO%.
41. ROCK MUSIC APPROACHES AT HIGH VELOCITY%.
42. NINE ADAM BAYE STUDY ON THE TWO PIECES%.
43. AN UNFRIENDLY DECAY CONVEYS THE OUTCOME%.
44. ABSTRACTION IS OFTEN ONE FLOOR ABOVE YOU%.
45. A PLAYED LADY RANKS ANY PUBLICIZED PREVIEW%.
46. HE TOLD US A VERY EXCITING ADVENTURE STORY%.
47. ON AUGUST TWENTY EIGTH%MARY PLAYS THE PIANO%.
48. INTO A CONTROLLER BEAMS A CONCRETE TERRORIST%.
49. I OFTEN SEE THE TIME ELEVEN ELEVEN ON CLOCKS%.
50. IT WAS GETTING DARK%AND WE WEREN'T THERE YET%.
51. AGAINST EVERY RHYME STARVES A CHORAL APPARATUS%.
52. EVERYONE WAS BUSY%SO I WENT TO THE MOVIE ALONE%.
53. I CHECKED TO MAKE SURE THAT HE WAS STILL ALIVE%.
54. A DOMINANT VEGETARIAN SHIES AWAY FROM THE G O P%.
55. JOE MADE THE SUGAR COOKIES%SUSAN DECORATED THEM%.
56. I WANT TO BUY A ONESIE%BUT KNOW IT WON'T SUIT ME%.
57. A FORMER OVERRIDE OF Q W E R T Y OUTSIDE THE POPE%.
58. F B I SAYS THAT C I A SAYS%I'LL STAY AWAY FROM IT%.
59. ANY CLIMBING DISH LISTENS TO A CUMBERSOME FORMULA%.
60. SHE WROTE HIM A LONG LETTER%BUT HE DIDN'T READ IT%.
61. DEAR%BEAUTY IS IN THE HEAT NOT PHYSICAL%I LOVE YOU%.
62. AN APPEAL ON JANUARY FIFTH DUPLICATES A SHARP QUEEN%.
63. A FAREWELL SOLOS ON MARCH TWENTY THIRD SHAKES NORTH%.

64. HE RAN OUT OF MONEY%SO HE HAD TO STOP PLAYING POKER%.
65. FOR EXAMPLE%A NEWSPAPER HAS ONLY REGIONAL DISTRIBUTION T%.
66. I CURRENTLY HAVE FOUR WINDOWS OPEN UP%AND I DON'T KNOW WHY%.
67. NEXT TO MY INDIRECT VOCAL DECLINES EVERY UNBEARABLE ACADEMIC%.
68. OPPOSITE HER SOUNDING BAG IS A M C'S CONFIGURED THOROUGHFARE%.
69. FROM APRIL EIGHTH TO THE PRESENT%I ONLY SMOKE FOUR CIGARETTES%.
70. I WILL NEVER BE THIS YOUNG AGAIN%EVER%OH DAMN%I JUST GOT OLDER%.
71. A GENEROUS CONTINUUM OF AMAZON DOT COM IS THE CONFLICTING WORKER%.
72. SHE ADVISED HIM TO COME BACK AT ONCE%THE WIFE LECTURES THE BLAST%.
73. A SONG CAN MAKE OR RUIN A PERSON'S DAY IF THEY LET IT GET TO THEM%.
74. SHE DID NOT CHEAT ON THE TEST%FOR IT WAS NOT THE RIGHT THING TO DO%.
75. HE SAID HE WAS NOT THERE YESTERDAY%HOWEVER%MANY PEOPLE SAW HIM THERE%.
76. SHOULD WE START CLASS NOW%OR SHOULD WE WAIT FOR EVERYONE TO GET HERE%?
77. IF PURPLE PEOPLE EATERS ARE REAL%WHERE DO THEY FIND PURPLE PEOPLE TO EAT%?
78. ON NOVEMBER EIGHTEENTH EIGHTEEN TWENTY ONE%A GLITTERING GEM IS NOT ENOUGH%.
79. A ROCKET FROM SPACE X INTERACTS WITH THE INDIVIDUAL BENEATH THE SOFT FLAW%.
80. MALLS ARE GREAT PLACES TO SHOP%I CAN FIND EVERYTHING I NEED UNDER ONE ROOF%.
81. I THINK I WILL BUY THE RED CAR%OR I WILL LEASE THE BLUE ONE%THE FAITH NESTS%.
82. ITALY IS MY FAVORITE COUNTRY%IN FACT%I PLAN TO SPEND TWO WEEKS THERE NEXT YEAR%.
83. I WOULD HAVE GOTTEN W W W DOT GOOGLE DOT COM%BUT MY ATTENDANCE WASN'T GOOD ENOUGH%.
84. NINETEEN TWENTY IS WHEN WE ARE UNIQUE TOGETHER UNTIL WE REALISE%WE ARE ALL THE SAME%.
85. MY MUM TRIES TO BE COOL BY SAYING H T T P COLON SLASH SLASH W W W B A I D U DOT COM%.
86. HE TURNED IN THE RESEARCH PAPER ON FRIDAY%OTHERWISE%HE EMAILED A S D F AT YAHOO DOT ORG%.
87. SHE WORKS TWO JOBS TO MAKE ENDS MEET%AT LEAST%THAT WAS HER REASON FOR NOT HAVING TIME TO JOIN US%.
88. A REMARKABLE WELL PROMOTES THE ALPHABET INTO THE ADJUSTED LUCK%THE DRESS DODGES ACROSS MY ASSAULT%.
89. A B C D E F G H I J K L M N O P Q R S T U V W X Y Z ONE TWO THREE FOUR FIVE SIX SEVEN EIGHT NINE TEN%.
90. ACROSS THE WASTE PERSISTS THE WRONG PACIFIER%THE WASHED PASSENGER PARADES UNDER THE INCORRECT COMPUTER%.
91. IF THE EASTER BUNNY AND THE TOOTH FAIRY HAD BABIES WOULD THEY TAKE YOUR TEETH AND LEAVE CHOCOLATE FOR YOU%?
92. SOMETIMES%ALL YOU NEED TO DO IS COMPLETELY MAKE AN ASS OF YOURSELF AND LAUGH IT OFF TO REALISE THAT LIFE ISN'T SO BAD AFTER ALL%.
93. SHE BORROWED THE BOOK FROM HIM MANY YEARS AGO AND HASN'T YET RETURNED IT%WHY WON'T THE DISTINGUISHING LOVE JUMP WITH THE JUVENILE%?
94. LAST FRIDAY IN THREE WEEK'S TIME I SAW A SPOTTED STRIPED BLUE WORM SHAKE HANDS WITH A LEGLESS LIZARD%THE LAKE IS A LONG WAY FROM HERE%.
95. I WAS VERY PROUD OF MY NICKNAME THROUGHOUT HIGH SCHOOL BUT TODAY%I COULDN'T BE ANY DIFFERENT TO WHAT MY NICKNAME WAS%THE METAL LUSTS%THE RANGING CAPTAIN CHARTERS THE LINK%.
96. I AM HAPPY TO TAKE YOUR DONATION%ANY AMOUNT WILL BE GREATLY APPRECIATED%THE WAVES WERE CRASHING ON THE SHORE%IT WAS A LOVELY SIGHT%THE PARADOX STICKS THIS BOWL ON TOP OF A SPONTANEOUS TEA%.
97. A PURPLE PIG AND A GREEN DONKEY FLEW A KITE IN THE MIDDLE OF THE NIGHT AND ENDED UP SUNBURNT%THE CONTAINED ERROR POSES AS A LOGICAL

TARGET%THE DIVORCE ATTACKS NEAR A MISSING DOOM%THE OPERA FINES THE DAILY EXAMINER INTO A MURDERER%.

98. AS THE MOST FAMOUS SINGLER-SONGWRITER%JAY CHOU GAVE A PERFECT PERFORMANCE IN BEIJING ON MAY TWENTY FOURTH%TWENTY FIFTH%AND TWENTY SIXTH TWENTY THREE ALL THE FANS THOUGHT HIGHLY OF HIM AND TOOK PRIDE IN HIM ALL THE TICKETS WERE SOLD OUT%.

99. IF YOU LIKE TUNA AND TOMATO SAUCE%TRY COMBINING THE TWO%IT'S REALLY NOT AS BAD AS IT SOUNDS%THE BODY MAY PERHAPS COMPENSATES FOR THE LOSS OF A TRUE METAPHYSICS%THE CLOCK WITHIN THIS BLOG AND THE CLOCK ON MY LAPTOP ARE ONE HOUR DIFFERENT FROM EACH OTHER%.

100. SOMEONE I KNOW RECENTLY COMBINED MAPLE SYRUP AND BUTTERED POPCORN THINKING IT WOULD TASTE LIKE CARAMEL POPCORN%IT DIDN'T AND THEY DON'T RECOMMEND ANYONE ELSE DO IT EITHER%THE GENTLEMAN MARCHES AROUND THE PRINCIPAL%THE DIVORCE ATTACKS NEAR A MISSING DOOM%THE COLOR MISPRINTS A CIRCULAR WORRY ACROSS THE CONTROVERSY%.

## D  15-SENTENCE TEST SET

The 15 sentences used to quantify the inference speed up in Section 5.2.1 are listed below (note that % corresponds to pause):

1. WHEN THE SUNLIGHT STRIKES RAINDROPS IN THE AIR%THEY ACT AS A PRISM AND FORM A RAINBOW%.
2. THESE TAKE THE SHAPE OF A LONG ROUND ARCH%WITH ITS PATH HIGH ABOVE%AND ITS TWO ENDS APPARENTLY BEYOND THE HORIZON%.
3. WHEN A MAN LOOKS FOR SOMETHING BEYOND HIS REACH%HIS FRIENDS SAY HE IS LOOKING FOR THE POT OF GOLD AT THE END OF THE RAINBOW%.
4. IF THE RED OF THE SECOND BOW FALLS UPON THE GREEN OF THE FIRST%THE RESULT IS TO GIVE A BOW WITH AN ABNORMALLY WIDE YELLOW BAND%.
5. THE ACTUAL PRIMARY RAINBOW OBSERVED IS SAID TO BE THE EFFECT OF SUPER IMPOSITION OF A NUMBER OF BOWS%.
6. THE DIFFERENCE IN THE RAINBOW DEPENDS CONSIDERABLY UPON THE SIZE OF THE DROPS%.
7. IN THIS PERSPECTIVE%WE HAVE REVIEWED SOME OF THE MANY WAYS IN WHICH NEUROSCIENCE HAS MADE FUNDAMENTAL CONTRIBUTIONS%.
8. IN ENHANCING AGENT CAPABILITIES%IT WILL BE IMPORTANT TO CONSIDER OTHER SALIENT PROPERTIES OF THIS PROCESS IN HUMANS%.
9. IN A WAY THAT COULD SUPPORT DISCOVERY OF SUBGOALS AND HIERARCHICAL PLANNING%.
10. DISTILLING INTELLIGENCE INTO AN ALGORITHMIC CONSTRUCT AND COMPARING IT TO THE HUMAN BRAIN MIGHT YIELD INSIGHTS%.
11. THE VAULT THAT WAS SEARCHED HAD IN FACT BEEN EMPTIED EARLIER THAT SAME DAY%.
12. ANT LIVES NEXT TO GRASSHOPPER%ANT SAYS%I LIKE TO WORK EVERY DAY%.
13. YOUR MEANS OF TRANSPORT FULFIL ECONOMIC REQUIREMENTS IN YOUR CHOSEN COUNTRY%.
14. SLEEP STILL FOGGED MY MIND AND ATTEMPTED TO FIGHT BACK THE PANIC%.
15. SUDDENLY%I SAW TWO FAST AND FURIOUS FEET DRIBBLING THE BALL TOWARDS MY GOAL%.

