# OpenReview forum: "Parallel Neural Text-to-Speech"
_ICLR.cc/2020/Conference — Reject_

### Official Review · AnonReviewer1 · 2019-10-23
**Official Blind Review #1**

**Rating:** 3

**Review:**

This submission belongs to the field of text-to-speech synthesis. In particular, it looks at the problem of efficient training and inference. Many current state-of-the-art approaches in the area employ an autoregressive component at one or another part of text-to-speech model making training and/or inference complicated/slow. The main idea behind this paper is to remove autoregressive components.

I believe there is a great deal of interest for efficient and fast text-to-speech. I believe stirring more interest towards non-autoregressive approaches is the right direction. I believe that the approach proposed does indeed accomplish the task of removing autoregressive components from the text-to-speech model. I find however that the results with parallel neural vocoders for the proposed approach to lag behind (autoregressive) Deep Voice 3 with the same set of vocoders even though when an autoregressive vocoder is used the situation is different.

I find that the presentation of this work to be lacking a balance. This work makes 2 contributions 1) non-autoregressive text-to-spectrogram and 2) non-autoregressive spectrogram-to-speech model.

 The first contribution is presented in a completely verbal manner to describe the complex process of text-to-speech mapping. For instance, I cannot find a proper technical description of 1) anywhere in the introduction, only statements that it is non-autoregressive and parallel.

The second contribution is squashed into a single page. Due to complexity involved I believe each of these contributions needs to be written (and assessed) separately. For instance, on page 6 you have a) encoder, b) decoder, c) VAE objective, d) STFT loss all discussed in very short details.

Other comments:
1) The Related work section would benefit from a bit more connected story than a list of RNN, flow and VAE based approaches.
2) Block diagrams are helpful but not having a mathematical description makes them more ambiguous than they should be.
3) "residual channel 256 conditioned", "miserable attention errors"
4) What is significance between 6 and 8 attention errors made by ParaNet and DV3?



**Experience Assessment:**

I have published in this field for several years.

**Review Assessment: Checking Correctness Of Derivations And Theory:**

I carefully checked the derivations and theory.

**Review Assessment: Checking Correctness Of Experiments:**

I carefully checked the experiments.

**Review Assessment: Thoroughness In Paper Reading:**

I read the paper thoroughly.

---

> ### Author Response · Authors · 2019-11-15
> **Response to Official Review #3**
>
> Thank you for the detailed comments and suggestions. Our response is as follows.
>
> 1, We organize the paper in a way that focuses more on building a fully parallel neural TTS system by proposing ParaNet and pairing it with various parallel neural vocoders, which also leads to an interesting comparison of vocoders. We focus less on WaveVAE because we treat it only as a separate vocoder option. We will add more details in the WaveVAE section in our final draft, although the main text will exceed the recommended 8 pages. We also consider putting more details in Appendix. We would like to hear your opinion about it. Many thanks for your suggestion.
>
> 2, We will make the related work section more connected, add more mathematical descriptions to diagrams, and modify the confusing descriptions in our final draft.
>
> 3, The 6 attention errors from ParaNet and 8 from DV3 are comparable. Note that, these results are obtained when using the attention mask at synthesis. In comparison, we showed that ParaNet has much fewer attention errors than DV3 (12 vs 37) without the attention mask.

---

### Official Review · AnonReviewer2 · 2019-10-24
**Official Blind Review #2**

**Rating:** 6

**Review:**

This paper proposes improvements to the TTS architecture in DeepVoice3.

For the most part (as far as I am aware), neural architectures for TTS are encoder-decoder based. The backbone can either be an RNN or a wavenet style model (autoregressive). Synthesizing audio from the feature representation (mel spectrogram) by a neural vocoder is usually an autoregressive model from the wavenet family.

In the current work, the main novelties seem to be the following:

1) Replace the autoregressive component in the decoder (seemingly inspired by DeepVoice3) with a non-autoregressive model. This could be very advantageous because synthesis in autoregressive models can be quite slow owing to the sample level generations (and to overcome this defect, faster sampling with inverse autoregressive flows has been used in parallel wavenet, clarinet, etc. and probability density distillation). Here, if I can interpret the paper correctly, the authors use a trained attention model (autoregressive), to distill attention for the non-autoregressive setup used, which would otherwise have difficulties learning alignment. The paper claims speed up of ~50X over DeepVoice 3 which is very significant.

2) In the mel to audio converter (vocoder), the proposal is a VAE Wavenet, with appropriate modifications for the sequence modeling. The authors mention that there are similarities to the approach used in Clarinet (this has a closed form KLD between the distilled distributions, which makes things easy).

Experiments: It is mentioned that poor attention alignments are the cause of many issues in these architectures (repeats, mispronunciations, skipped words, etc.), and they go on to show that their architecture fares well as regards these metrics, while maintaining a comparable MOS score with DeepVoice3+Wavenet.

Evaluations: I am reasonably convinced by the audio demos provided.

My thoughts:
The paper is generally a good addition to the TTS literature. These are difficult to implement, brittle setups, and a practitioner could spend a lot of time debugging their broken attention curves. To that end, I feel that any technique that throws light into the modeling process would be useful for the practitioner. It is suggested that we can use a non-autoregressive model with speedups. Likewise, they also use a wavenet VAE, which they claim can be trained without distillation (could the authors please clarify this point?)

I do, however, feel that the paper has a few drawbacks.

1) The presentation is not at all clear. This is a very subjective comment, but I feel that this work might be unreadable to someone who hasn't studied the literature (starting from Tacotron, DeepVoice 1, 2, 3, wavenet, parallel wavenet, transformer, distillation, etc.). Furthermore, the paper does not seem to make things any clearer with succinct architecture diagrams. Just as a comparison, I would like to draw attention to Tacotron (1, 2) in which I think the details can actually be worked out with some effort.

2) Why are we using the WaveVAE - is this just to do away with the probability distillation in wavenet? Are we also using the  IAF setup as described in Kingma's work?

3) I really feel that we need more architecture diagrams for this work to be useful. That being said, the authors do provide a list of hyperparameters for potential use.

4) Distilling attention seems to require a previously trained autoregressive attention model. What if we don't have one ready at hand?

In summary, while I see that the work will definitely be useful to the Speech Synthesis practitioner, the clarity of the paper could be improved and we need a few more diagrams (maybe even code) to make it implementable.

**Experience Assessment:**

I have published one or two papers in this area.

**Review Assessment: Checking Correctness Of Derivations And Theory:**

N/A

**Review Assessment: Checking Correctness Of Experiments:**

I assessed the sensibility of the experiments.

**Review Assessment: Thoroughness In Paper Reading:**

I read the paper at least twice and used my best judgement in assessing the paper.

---

> ### Author Response · Authors · 2019-11-15
> **Response to Official Review #2**
>
> Many thanks for your detailed review. We also appreciate your interests in our work. Our response is as follows.
>
> 1.  The motivation for WaveVAE is that the model can be trained from scratch without requiring a two-stage knowledge distillation (pre-training an autoregressive teacher model and then distilling a student model from it). That means the encoder and decoder in WaveVAE will be trained jointly which simplifies the training process. We use IAF as the decoder in WaveVAE.
>
> 2. We will clarify our presentation, give more background information in the related work section, and add more descriptive illustration in diagrams (also more diagrams in appendix) in the final version.  Thank you so much for your suggestion.
>
> 3. We did ablation studies and found that ParaNet failed to learn meaningful attention alignment without distillation from a pretrained autoregressive attention model. That means if we don’t have one at hand, we will need an alignment model (e.g., based on force alignment) to guide the learning of ParaNet.

---

### Official Review · AnonReviewer3 · 2019-10-28
**Official Blind Review #3**

**Rating:** 1

**Review:**

This paper describes work on Parallel Neural Text-To-Speech.

There are a number of efforts in this direction going on simultaneously.  FastSpeech is one that was noted in public comment.  I have seen other work under submission as well.  I don't believe this undermines the fundamentals of the work, but it will color claims of being "first" by each submission.

WaveVAE is an interesting approach to a parallel neural vocoding.  However it performs works than the IAF distilled ClariNet parallel vocoder with the same amount of parameters.  It is not clear what advantages WaveVAE has over ClariNet.

More problematic, ParaNet is described as providing a speedup over Deep Voice while maintaining quality.  However, this claim is somewhat misleading.  The maintained quality is only achieved when using an autoregressive wavenet vocoder.  It appears as though the inference speed up is only measured on the feature to mel conversion, while the wall-clock inference time is (likely) dominated by wavenet inference (a notoriously computationally intensive process).  When evaluated with parallel vocoders (ClariNet, WaveVAE or WaveGlow) ParaNet performs quite poorly when compared to Deep Voice 3. (This is especially true when using WaveGlow -- the best performing vocoder for DV3 MOS=3.96, ParaNet MOS=3.21).  While there are fewer attention errors (cf. Table 2) this doesn't seem to impact MOS to a great degree.

Attention Distillation is somewhat equivalent to using an external alignment or duration model as is done in classical TTS.  Requiring this external resource somewhat undermines the claim of being an "end-to-end" TTS system.  It is worth comparing attention distillation to, say, training a traditional alignment model based on forced alignment and feeding the target durations as a conditioning feature along with the text.  This is what ParallelWaveNet (https://arxiv.org/pdf/1711.10433.pdf) does.  From that perspective it is not clear why ParaNet+neural vocoder should be considered as "more parallel" than ParallelWaveNet or other IAF vocoding approaches paired with a traditional front end.

**Experience Assessment:**

I have published one or two papers in this area.

**Review Assessment: Checking Correctness Of Derivations And Theory:**

I assessed the sensibility of the derivations and theory.

**Review Assessment: Checking Correctness Of Experiments:**

I carefully checked the experiments.

**Review Assessment: Thoroughness In Paper Reading:**

I read the paper thoroughly.

---

> ### Author Response · Authors · 2019-11-15
> **Response to Official Review #3**
>
> Thank you for the comments and suggestions. Our response is as follows.
>
> 1. We agree that the simultaneous work on the same topic doesn’t undermines the fundamentals of each work. Indeed, it demonstrates the wide interest & practical importance of the topic. We didn't discuss the simultaneous work in related work, because it would encourage searching on arXiv (e.g., for checking the simultaneousness) and hurt anonymous review process. We would like to discuss them in the final draft.
>
> 2. The main advantage of WaveVAE over ClariNet is that WaveVAE can be trained from scratch without knowledge distillation, which largely simplifies both the training process and the model development. In comparison, ClariNet requires a two-stage training process, where the authors first pretrain an autoregressive teacher model and then distill a student model from it.  Although WaveVAE still performs worse than ClariNet vocoder, we believe it is an interesting endeavor.  In particular, it is very challenging to model raw audio with VAE using non-autoregressive decoder because of the over-smoothing issue.  The vanilla implementation couldn't even produce intelligible speech.
>
> 3.  We tried a long time tuning the WaveGlow model on our internal dataset. But there are still constant high-frequency noises present in the synthesized speech. To remedy this, we applied the denoising function with strength 0.1 in the open source repository. The denoising function is effective when the predicted mel spectrograms are from DV3, but not quite effective when the predicted mel spectrograms are from ParaNet. These noises have a huge impact on the MOS evaluation results, which is the major reasons for the MOS score gap between DV3 + WaveGlow and ParaNet + WaveGlow.
>
> 4.  We didn’t claim the proposed model is “end-to-end” in our paper. Even without the attention distillation, it is still not end-to-end training, because it involves the separate training of text-to-spectrogram model and neural vocoder. But we agree that the comparison between traditional alignment model and attention distillation could be interesting.
> Note that, the traditional front end usually uses RNN or autoregressive model to predict phoneme duration and fundamental frequency, thus it is not parallel at frame-level (e.g., typically ~200 frames per second) during synthesis.  In contrast, ParaNet generates spectrogram frames in parallel, and feed them to parallel IAF vocoder. Thus, ParaNet + IAF vocoder should be considered as “more parallel” than Parallel WaveNet.

---

### Public Comment · ~Rafael_Valle1 · 2019-10-07
**Related work**

Thank you for sharing your work with our community.

I'm writing to call attention to the work by Yi Ren et al. published under "FastSpeech: Fast, Robust and Controllable Text to Speech" in May 2019. In their paper, they describe a model for parallel mel-spectrogram generation. Below is their abstract:

"""Neural network based end-to-end text to speech (TTS) has significantly improved the quality of synthesized speech. Prominent methods (e.g., Tacotron 2) usually first generate mel-spectrogram from text, and then synthesize speech from mel-spectrogram using vocoder such as WaveNet. Compared with traditional concatenative and statistical parametric approaches, neural network based end-to-end models suffer from slow inference speed, and the synthesized speech is usually not robust (i.e., some words are skipped or repeated) and lack of controllability (voice speed or prosody control). In this work, we propose a novel feed-forward network based on Transformer to generate mel-spectrogram in parallel for TTS. Specifically, we extract attention alignments from an encoder-decoder based teacher model for phoneme duration prediction, which is used by a length regulator to expand the source phoneme sequence to match the length of target mel-spectrogram sequence for parallel mel-spectrogram generation. Experiments on the LJSpeech dataset show that our parallel model matches autoregressive models in terms of speech quality, nearly eliminates the problem of word skipping and repeating in particularly hard cases, and can adjust voice speed smoothly. Most importantly, compared with autoregressive Transformer TTS, our model speeds up the mel-spectrogram generation by 270x and the end-to-end speech synthesis by 38x."""

---

> ### Author Response · Authors · 2019-10-08
> **Thanks for your comment**
>
> Hi Rafael,  thanks for your comment. We have noticed that work.  Note that, our work was done in parallel with that paper as indicated by arXiv submission history. We didn't discuss it in related work, because it would encourage people to search the submission on arXiv. Since you raise it up, we would like to clarify it here. We would like to reference it in the final version.

---

### Decision · Program_Chairs · 2019-12-19

**Decision:**

Reject

**Comment:**

The paper proposed a non-autoregressive attention based encoder-decoder model for text-to-sepectrogram using attention distillation. It is shown to bring good speedup to conventional autoregressive ones. The paper further adopted VAE for the vocoder training which trains from scratch although performs worse than existing method (e.g. ClariNet).

The main concerns for this paper come from the unclear presentation:
* As the reviewer pointed out, there're some misleading claims that the speedup gains was obtained without the consideration of the full context (i.e. not including the whole inference time).
* The paper failed to clear present the architectures developed/used in the paper and the differences from those used in the literature. The reviewers suggested the use of diagram to aid the presentation.
* The two contributions are unbalanced presented. Due to the complexities involved, it's better to explain things in more details.
The authors acknowledged the reviewers comments during rebuttal, but did not make any changes to the paper.